# Advanced Technologies and Their Use in Smart City Management

Josef Vodák, Dominika Šulyová * and Milan Kubina

Faculty of Management Science and Informatics, University of Zilina, Univerzitna 8215/1, 010 26 Zilina, Slovakia; josef.vodak@fri.uniza.sk (J.V.); milan.kubina@fri.uniza.sk (M.K.)
* Correspondence: dominika.sulyova@fri.uniza.sk; Tel.: +421-41-513-4022

**Abstract:** Building Smart City management concepts is based on the implementation and use of advanced technologies. The primary impulse for writing the article was the ambition to identify the current advanced technologies of Smart City management. The aim of the article is to propose a general model for the implementation of advanced technologies for Smart City management, based on the knowledge gained from the analysis of literature and case studies. In order to fulfill the set goal, it is necessary to obtain answers to two research questions. The findings were obtained through a secondary analysis of the literature, i.e., relevant articles from the scientific databases Web of Science and Scopus analysis of case studies of the best Smart Cities practices. According to the Smart City Index 2020 and IESE Cities in Motion, the leaders among the Smart Cities are Singapore and London, followed by Helsinki. In addition to the analyses, the article also uses methods of summarization, comparison, creativity, logic, induction and deduction. Smart Cities use 12 identified advanced technologies in their practice. Strategic management in Singapore, London and Helsinki adapts technology to the needs and requirements of its citizens, thus connecting the technological aspect with the managerial and social aspects. The contributions of the work include results for fellow researchers and a model for strategic management of new Smart Cities. The results of the article provide fellow researchers with the findings of a secondary analysis of relevant articles, from which they can draw when writing their own publications without the need for time-consuming search of the articles about this topic in databases. The general model of implementation of advanced technologies serves as a basis for strategic management of new Smart Cities that want to implement a technological base and at the same time do not want to forget the managerial and social aspects. Testing the model in practice with a new Slovak Smart City is part of future research activities.

**Keywords:** advanced technologies; Smart City; UAV; management; internet of things

## 1. Introduction

An essential part of building a developed knowledge society is the effective use of advanced technologies (intelligent applications, process digitization, use of UAV resources in combination with information and communication technologies (ICT), real-time data processing and evaluation) in process management of large and complex systems such as a Smart City. Here, the implementation of these technologies is effectively used in the processes of city management, managing limited resources and addressing stakeholder requirements [1,2]. Information and communication technologies contribute to the promotion of the sustainability of the Smart City and a higher quality of life of its citizens. The implementation of sensors, action units, communication platforms and interconnected systems generates data architectures [3]. The Smart City concept combines two current trends—the digital revolution and urbanization [2]. Information and communication technologies are being developed in a particular social and cultural environment in collaboration with communities, which are the driving force behind innovation and sustainable development [4]. Dyer, Gleeson and Gray argue that in order to secure urban management processes, also

referred to as collaborative urbanism, it is not enough just to implement advanced technologies. It is important that citizens connect the current culture of participation in Smart City projects with the use of the Internet [5,6].

According to Simonofski et al., Smart City represents not only a technological but also a social system. If the strategic management of the city does not create a strategy that will reflect the values and expectations of the inhabitants, a frequent consequence is the failure to achieve the set goals. Residents participate the form of a social element to the concepts of Smart City act as democratic participants, co-creators and users of advanced technologies [7]. Democratic participants participate in setting the goals and priorities of Smart City strategies and projects, creating communities and cooperation activities. Interaction with co-creators is mediated through online platforms that are centrist-oriented to the needs and values of citizens. Advanced technology users share data on a daily basis, which is collected, distributed and analyzed through advanced technologies and then used to support management and decision making [7].

According to Allahar, the number of world Smart Cities will increase by an additional 88 by 2025. In his article, Allahar presents the current problem of Smart Cities, namely the ambiguity in defining specific technologies that should be implemented in Smart City concepts [8]. Based on the above findings, the authors are of the opinion that it is necessary to look for new solutions that will form a model for strategic management of cities that plan to start building Smart City concepts that will reflect not only the technological but also the social requirements of residents. The purpose of the article is to identify a unique set of technologies that can be used to manage a Smart City. The aim of the article is based on the knowledge gained from the analysis of the literature and case studies to propose a general model for the implementation of advanced technologies for the management of a new Smart City. To fulfill the aim of the article, two research questions were selected:

- How are the selected advanced technologies for Smart City management currently used in the world's best practice?
- How is it appropriate to implement advanced technologies for Smart City management in general?

The fulfillment of the aim of the article was realized through a secondary analysis of the literature, which served to identify the key technologies used in the management of a Smart City. The analysis of current strategies is based on the findings of the literature analysis and provided a description and comparison of three best practice cases of Smart Cities—Singapore, London and Helsinki. The list of summarized technologies found in Section 1 was used for the design of the own general model and confirmed in practice in Section 3. In this way, relevant answers to the identified research questions were obtained. The limitations of research were formed by national and cultural differences of cities. A limitation is also the adaptation of the model to the specific conditions of the selected city or the absence of its testing, which is planned in the future. The article provides a summary of information on the issue for fellow researchers and the general model represents a proposal for the implementation of advanced technologies for the strategic management of emerging Smart Cities.

In the secondary analysis of the literature, technologies were described according to the results of Marr and Choudhera [9,10]. According to the Marra survey in 2018, the advanced technologies in the Smart City concept are used mainly in the areas of building cooperation with citizens, security, transport and delivery of goods, and efficient use of limited resources [9]. Choudhery is of the opinion that the key technologies in managing a Smart City are the Internet of Things, Big Data, information and communication technologies (online platforms, dashboards and systems supporting citizens' trust in technology), UAVs and blockchain [10].

Based on data from Marr and Choudhera, it is possible to identify 12 key technologies for effective management of a Smart City. For the area of building collaboration, it is necessary to collect data from the field through the Internet of Things, which operates on the basis of the Internet network. The data in the form of Big data are then analyzed. Their

voluntary sharing is determined by trust in systems and the level of security. Strengthening citizens' participation also depends on transparent information being provided through online platforms, dashboards and models. The area of transport and resource use efficiency is currently covered by the trend of unmanned aerial vehicles [9,10].

## 2. Research Background

### 2.1. Internet of Things in the Smart City Concept

The impact of Internet of Things technology is reflected in the implementation of smart solutions in urban agglomerations through new forms of infrastructures and processes, efficient use of limited resources, security monitoring through activity maps, cooperation in service provision and strategic city management. The collected data simplify capacity management, decision making and optimization [11]. The connection is implemented on the basis of sensors, action units, the Internet and mobile applications, i.e., the basic infrastructure. Internet of Things technology simplifies research work, cooperation and brings integrated solutions to problems related to the field of smart cities in connection with their management [12,13]. Selected definitions of the term Internet of Things can be found in Table 1.

**Table 1.** Defining the term IoT (Internet of Things) in the Smart City concept.

| Authors | Definition |
|---|---|
| KPMG, 2019 [11] | IoT is a modern evolution for Smart Things, the environment and interaction. It builds a strong foundation for economic growth and a higher quality of life. The primary purpose of this technology is to collect, analyze and maximize data. |
| Mao, 2019 [14] | IoT technology in Smart Cities is used for process monitoring, communication between devices over the network, and represents a resource and opportunity for proper management of the city and the safety of its inhabitants. |
| Jones, 2020 [15] | IoT has a huge impact on life, the planet and all the people in it. It creates an active component that has a positive effect on efficiency, sustainability and the development of smart cities. |
| Meola, 2020 [16] | Smart Cities use the Internet of Things in the form of sensors, action units and connected facilities so that city management can use these data to plan, manage and measure their activities well. |

Areas of use of the Internet of Things are [17]:

- Intelligent lighting.
- Real-time monitoring—energy consumption, water consumption, air quality.
- Waste management.
- Transport—parking, smart intersections.
- Intelligent services—security, education, healthcare.

The specificity of each of the Smart Cities requires the implementation of such IoT technology models that will reflect the size of the current population in a particular city [17]. Large cities with population higher than 1,000,000 focus on urban infrastructure management (transport, security). Medium size cities have approx. 500,000 inhabitants, they typically focus on increasing the efficiency of waste management and use of limited resources. Small cities with 100,000 and fewer need to improve the quality of life with associated protection of limited resources [17].

Currently, it is a widespread trend to interconnect multiple devices, which is the essence of IoT. The future trend that follows is IoE, i.e., Internet of Everything. Under that concept, all elements of people, data, processes and things will be connected [18]. The technical sensors of the Internet of Things are divided into six groups [19].

Environmental sensors monitor sources and state of the environment. Mobile sensors collect data from the environment. Ubiquitous sensors are based on radio frequency technology (RFID). Remote sensors can be satellite or terrestrial ones. Collective sensors include online networks, mobile phones, media. People as sensors are measuring the level of stress, heat, they provide voluntary geographical information, the so-called VGI [19,20].

### 2.2. Internet of Things and Big Data

The development of Internet of Things and Internet of Everything technologies is creating a new era of so-called dataism, which is perceived by Smart Cities as a comprehensive data processing system. The Big Data flow mediates information between the public and private sectors, thus supporting informatization, digitization and stakeholder competitiveness. The key processes for the effective connection of the Internet of Things and management are monitoring, diagnostics (research, statistics), detection of deviations and their elimination in the phase of smart control, automation of solutions through applications and efficient use of resources. Data from sensors are transmitted over the network to database repositories, from which they are then distributed for analysis, management and decision support and optimization of smart applications [21]. Current strategic management cannot process Big Data efficiently. In practice, the following five changes are required [18,22]:

- New operating models which should be bottom-up, centrist-oriented;
- Sustainable development plans with synergies between resources, sensors and functions;
- Government data ownership where value will be generated by sharing public and private sector data to better support governance and decision making that reflects the interests of all stakeholders;
- New management models based on collaboration and the use of trendy technologies;
- Social change because environmental challenges affect the economic and social side of the city, which must be effectively managed through the management of limited resources in order to eliminate negative impacts on the environment, sustainability and future generations.

The biggest shortcoming of current cities is not the lack of data from the monitoring process, but the absence of their processing, analysis and use. The output of the new systems will be the trend of IoE, which will contribute to the creation of a new value—information freedom. However, maximum data flow and interconnection of all elements reduces the degree of individuality, protection of personal data, autonomy and privacy. The consequence of technology development is strategic management focused on the processes of decision making and selection of relevant information [22].

### 2.3. T-ARS, SDL, SDN and WSN Systems

A necessary element of building Smart City concepts is trust, which is supported in intelligent systems on the basis of advanced technology of the Internet of Things, Big Data and information and communication technologies, i.e., T-ARS (trust-aware systems). The operation of systems that have a positive effect on building trust is based on D2D (device-to-device) devices, user ranking, prediction and subsequent evaluation [23].

The Smart City Service System (SSSC) connects the urban concept with the principles of SDL (Service-Dominant Logic). A holistic view of the modern approach includes [24]:

- data collection via sensors;
- information integration;
- knowledge database, i.e., comprehensive analysis, visualization, optimization and modeling of acquired information for the needs of management and decision making of strategic management of Smart Cities.

The integration of Cloud computing, the Internet of Things and Big Data analysis is mediated through SDN (Software-Defined Network). The main goal of the Software-Defined Network is to create a vision of a Smart City based on a unified communication infrastructure of stakeholders [25]. Centralized system control creates a secure user environment in real time. The Software-Defined Network (SDN) has three levels. Data infrastructure composed of interconnected components based on established standards and rules. Control levels based on a centralized control system via a network operating system (NOS). Application is based on operations and functions through applications and platforms [3].

According to the National Institute of Standards and Technology (NIST), it is essential to build secure communication platforms that use the Software-Defined Network (SDN) in conjunction with ICT (information and communication technology). A modern intelligent infrastructure consists of interconnected devices communicating over a network (so-called device-to-device), Wi-Fi (Wireless Fidelity) connection, data processing and their storage in data centers, e.g., Cloud [3].

The widespread technology of the Internet of Things is the so-called Wireless Sensor Network (WSN). The sensor network forms a virtual layer for data transmission. IoT-WSN (Internet of Things-Wireless Sensor Network) is composed of a huge number of sensors and nodes that are located on moving objects in a large area of a Smart City. Technology is used primarily in the process of monitoring, management and decision making [25].

### 2.4. Modern City Models, Information Panels and Networks

Solving the city's problems through electronic public administration (eGov) is based on the principle of the Smart City website. A new model for their creation is MEPA (Urban Model of the Access Platform), which focuses on [26]:

- transparent information;
- citizen involvement;
- inclusion, access and elimination of differences;
- sustainability;
- the values of decentralization of power and accountability.

Companies such as IBM, Cisco and Accenture play the role of a supplier of modern and advanced technologies for the Smart City concept [27]. City dashboards have evolved to be able to aggregate Big Data in real time and easily visualize it. Online data processing (OLAP) covers key performance indicators, transport, weather, limited resources (water, energy, air, etc.), health and industry. The connection is mediated via the classic Internet protocol TCP/IP (Transmission Control Protocol/Internet Protocol). The architecture of information panels contains three layers. The data layer collects data, stores and monitors data via API (Application Programming Interface) libraries and user interface, i.e., GUI. The application layer manages data in the central layer of the architecture, analyzes indicators and connects the data and presentation layer. The presentation layer is used to communicate with residents and displays information about services [28].

The increase in the use of intelligent mobile devices has led to the creation of the MEC network (mobile edge computing). This advanced and modern technology is based on endpoints that can store data with low power consumption. Safety is mediated through UAV (unmanned aerial vehicle) devices, i.e., drones that monitor data transmission [29].

### 2.5. Security in Smart City

According to Chang Wu, Sun and Jim Wu, the trend in security management in the Smart City concept is the ISP (information security policy) model. The basic elements of the model are asset management, risk, auditing and continuous improvement of urban processes, cooperation with business partners and awareness raising through education and training [30].

Future challenges for the security of Smart City concepts are to implement effective preventive strategies for the development of IoT networks, to prefer data storage in Fog, to strengthen personal data protection, to minimize the volume of data shared via IoT, to reduce security costs, to create a unified concept Smart City architecture. IoT architecture for Smart Cities has an access layer (sensors), network (Wi-Fi), data storage (Cloud, Fog computing) and application. The connecting elements are data management and security [31].

### 2.6. Unmanned Aerial Vehicle (UAV)

UAVs are mobile platforms with built-in microprocessors, sensors, wireless communication devices and cameras. They were initially used for a defined purpose by the US

Department of Defense. They are currently being implemented into Smart Cities variable area concepts, thus contributing to the fulfillment of set goals [32,33]. UAVs were primarily used in the military field, and currently monitor and perform services for healthcare, mobility, supply of goods and security. In this way, they contribute to the development of the city's critical infrastructure. The benefits of unmanned aerial vehicle equipment are higher productivity, lower costs, flexibility and a high level of repeatability. The competitive advantage of mobile drones is the data transfer rate, which is key for Smart City concepts. Outdated data lose their useful value. For maximum effect of an unmanned aerial vehicle, it is necessary to optimize the flight trajectory, taking into account the maximum number of IoT devices in the network [32,34]. There are nine areas of application of the UAV in Smart Cities [32]:

- Traffic monitoring and control for prediction and elimination of traffic jams in real time, search for a parking place, creation of traffic simulation models;
- Healthcare—transport of defibrillators, medical supplies, unmanned aerial vehicle as an ambulance;
- Crowd monitoring, for example, police in the UK have used a drone to detect a car thief; an unmanned aerial vehicle uses abnormal motion detection or video streaming to track a crowd;
- Critical Infrastructure Inspection for scanning problem areas for inspection of bridges, pipelines, power lines and buildings;
- Management of limited resources as support for agriculture, water distribution, monitoring of emissions, pollution;
- Tourism with drone as a virtual guide or camera;
- Geodetic works for remote sensing, creation of 3D models, support of geographic information systems;
- Prediction and solution of natural disasters, such as fires, earthquakes, floods;
- Delivery of goods, wireless communication and unmanned aerial vehicle taxi function or data collection on the principle of drones and WSN (Wireless Sensor Network).

Photogrammetry uses drones to convey information about earthquakes, volcanic or flood activity, detection of green areas, etc. [35]. UAVs are suitable for use if they are to perform a task that is dangerous to humans, or if they can perform it more efficiently. In Israel and Australia, drones map surface water reserves [33]. The intelligent platform for drones contains the following four components [36]: the cloud server which stores data, coordinates tasks, communicates with unmanned aerial vehicle devices; the operator who is a specialist who can manually control the drone if necessary; the client who requests a service; and the work platform, which performs tasks, and responds to the platform and operator commands [36].

### 2.7. Technologies Used in Monitoring Limited Resources

Caragliu et al., in 2011, argued that the successful Smart City should "manage investments in human capital, technological innovation, and limited resource management," [37]. Currently, the technical focus and perception of the Smart City concept prevail. However, according to Raharjana, it is also important to take into account environmental aspects, especially the management of limited resources [38].

According to a scientific article by Ozkay and Erdin from 2020, 37% of the population consider it critical to effectively manage limited resources, 35% prefer to eliminate pollution, 22% want to increase the attractiveness of the region's natural conditions and 6% give the highest priority to biodiversity protection [39].

Smart environmental monitoring includes measurement of noise, pollution, temperature, weather, regulation of limited resources (e.g., water), using radar systems, fire detection, waste management, solar energy, LCD/LED (liquid crystal display/light-emitting diode) screens or early warning systems [40]. The basic element of smart monitoring are sensors that collect data in the field; afterwards, these are transferred to analytical centers and comprehensively explored. The final results from the field are compared with the plans

and standards. If the values are exceeded, corrective action is taken. The main advantage is prediction through early warning systems [40].

Iftikhar et al., in 2019, proposed a system model of data sharing in the Smart City concept, which can also be applied to the environment. The endpoints are connected to IoT technology, which transmits the data to a specific database called Blockchain. The benefits are operational changes of data in chronological order, support for the social dimension of Smart City, decentralization and greater data security [41–43].

### 2.8. Summary of a Set of Identified Advanced Technologies

Based on the secondary analysis of the literature in Sections 2.1–2.7, it was possible to identify 12 commonly used advanced technologies that serve to manage Smart City concepts, namely:

- Internet of Things (IoT);
- Big Data;
- T-ARS system (trust-aware systems);
- SDL (Service-Dominant Logic);
- SDN (Software-Defined Network);
- WSN (Wireless Sensor Network);
- MEPA model (urban model of the access platform);
- Dashboards;
- MEC (mobile edge computing) network;
- ISP (information security policy) security model;
- UAV equipment (unmanned aerial vehicle);
- sensors for environmental monitoring.

A set of 12 advanced technologies forms the necessary input for the analysis of case studies in Section 4 and the design of our model in Section 5.

## 3. Materials and Methods

### 3.1. Searching Strategy

Scientific articles and studies that formed the basis for the analysis of the literature in Sections 2.1–2.7 were searched for in the relevant databases as Web of Science and Scopus. The strategy of searching for relevant articles consisted of 5 steps: search for articles based on selected keywords, selection based on abstract availability criteria, open access and elimination of duplicate articles. The selection was made by filtering based on the following keywords in Tables 2 and 3.

**Table 2.** Number of relevant articles in databases Web of Science and Scopus in 2016–2021—part one.

| Year | Key Words "Information and Communication Technologies", "Governance of Smart City" | |
|:---:|:---:|:---:|
| | Web of Science | Scopus |
| 2021 | 4 | 20 |
| 2020 | 23 | 42 |
| 2019 | 13 | 52 |
| 2018 | 8 | 28 |
| 2017 | 16 | 30 |
| 2016 | 5 | 24 |

Source: own processing according to results in Web of Science and Scopus databases.

In the Scopus database, articles were filtered using two search strings:

TITLE-ABS-KEY (information AND communication AND technologies AND used AND for AND governance AND smart AND city)

TITLE-ABS-KEY (uav AND in AND smart AND cities AND resources)

**Table 3.** Number of relevant articles in databases Web of Science and Scopus in 2016–2021—part two.

| Year | Key Words "UAV (Unmanned Aerial Vehicle)", "Smart Cities", "Potential Resources" | |
|---|---|---|
| | Web of Science | Scopus |
| 2021 | 4 | 8 |
| 2020 | 8 | 13 |
| 2019 | 5 | 12 |
| 2018 | 3 | 8 |
| 2017 | 2 | 2 |
| 2016 | 2 | 3 |

Source: own processing according to results in Web of Science and Scopus databases.

The first filter generated 265 articles (69 in Web of Science, 196 in Scopus, Table 2) the second filter focused on unmanned aerial vehicle yielded 70 relevant articles (24 in Web of Science, 46 in Scopus, Table 3).

The second filtering was based on relevance criteria, i.e., articles should not be duplicates and had to include an abstract. After the second selection, 44 resources remained in the first part on ICT (information and communication technologies), and 8 scientific articles on UAVs (unmanned aerial vehicles).

The third filtering had to satisfy a simple search for the article on the Internet, i.e., the article was to be accessible via open access. For some information and communication technologies, 21 articles were discarded in this way, for an unmanned aerial vehicle only 2 were discarded.

In the fourth phase of filtering, the abstract of each article was analyzed and its suitability and relevance were assessed. The authors used 23 relevant articles about the use of information and communication technologies (ICT) in the management of Smart City and limited resources and 6 articles focused on unmanned aerial vehicle (UAV) in Smart Cities and limited resources.

In addition to the secondary analysis of the literature, the article also used as input the analysis of case studies based on secondary data, methods of summarization and comparison (Table 5), and logic and creativity in creating authors' own general model of implementing advanced technologies for Smart City management (Section 5), including deduction (Section 5) and induction (Section 6).

*3.2. Procedure for Secondary Analysis of Case Studies*

For the purpose of Section 4, i.e., secondary analysis of case studies, the goal was to confirm the use of 12 identified technologies in the world's best practice and thus contribute to solving the problem in the form of ambiguous identification of technologies that should be implemented by the emerging Smart Cities.

In order to fulfill the set purpose, it is necessary to obtain an answer to the following research question:

- How are selected advanced technologies for Smart City management currently used in the world's best practice?

Smart Cities were selected based on their ranking in the Smart City Index 2020 and IESE Cities in Motion Index 2020 rankings, the leading results of which are in Table 4.

**Table 4.** Ranking Smart Cities according to world indices.

| Ranking | Index | |
|---|---|---|
| | Smart City Index 2020 | IESE Cities in Motion Index 2020 |
| 1 | Singapore | London |
| 2 | Helsinki | New York |

Source: own processing according to [44–46].

The Smart City Index and the IESE Index were selected on the basis of temporal and thematic relevance, as they reflect the current results in several indicators, which are the priority indicators of city's technological maturity. The first selection was the best practice of rankings, i.e., Singapore and London.

As the comparison of 2 places is not enough to ensure objective results, it was necessary to make a second selection, including Helsinki and New York, ranked second in the indices (Table 4). When choosing the third place, the preference for locality in Europe was taken into account, therefore the city of Helsinki was selected. Data on 12 identified advanced technologies were searched for in selected cities (Section 2.8.). The current strategies of Singapore, London and Helsinki were used as a basis. The case studies focused practically on the use of information and communication technologies and unmanned aerial vehicles as a function of modern and advanced technologies. They should meet the condition of temporal relevance, i.e., year 2019, 2020 and 2021. The effect of comparative analysis was to mediate a unified set of advanced technologies that should be implemented by the newly emerging Smart Cities into their concepts. A specific description, summary and comparison of the use of advanced technologies in selected cities can be found in Section 4. The limitation was access to strategies as only publicly available information on the Internet could be used.

*3.3. Model Creation Procedure*

After confirming the use of a defined set of 12 technologies in practice and identifying the practical way of their use in Singapore, Helsinki and London, it was necessary to answer the second research question as follows:

- How is it appropriate to implement advanced technologies for Smart City management in general?

The answer to the question and the actual solution to the problem is our own general model of implementing advanced technologies used in Smart City management. The model was constructed based on findings from the secondary analysis of the literature (Section 2) and the results of case studies (Section 4), including the method of logic and creativity. The limitations of research were formed by national and cultural differences of cities. A limitation is also the adaptation of the model to the specific conditions of the selected city or the absence of its testing, which is planned in the future.

## 4. Results

This section presents data obtained by secondary analysis of best practice strategies from Singapore, London and Helsinki, including comparison and key findings.

*4.1. Secondary Analysis*

4.1.1. Singapore

Singapore began developing a digitization plan between 1980 and 1990. The primary goal was to create technologies that could be used by the city's elderly. Intelligent management of Smart Cities uses basic IT (information technology) infrastructure, IoT (Internet of Things), sensors, Big Data, WSN (Wireless Sensor Network). The result is data integration, improved processes in the city and the provision of quality services for stakeholders. An important Smart City Singapore development includes six projects [47,48].

National Digital Identity (NDI) represents secure and simple transactions between the public and private sectors. Citizens can use auto-from filling via the intelligent MyInfo platform; verification takes place by credit card and bank account number. Smart Urban Mobility is a method of electronic payment for tickets, searching for free parking space via sensors and applications, efficient transport in terms of time and place. Sensor platform for collection and subsequent analysis of data for the needs of management processes in the city. Sensors monitor air quality, environmental pollution, watercourses, flood risk, water and energy consumption, transport activities, etc. Communication platforms contain data on the birth of a child which are registered online in Singapore; seniors can obtain

relevant information about the benefits of Smart City to their lives and communicate with the government through the Moments of Life Initiative. Core Operations Development Environment and eXchange (CODEX) is a digital platform that offers citizens fast and secure service delivery. Electronic payments which take place via a mobile number and individual identification of the citizen, the so-called Unique Entity Number [47,48].

The management of Smart City Singapore has elements of digitization at the heart of development. Transformation has generated new strategies, policies, processes and organizational structures. A key element of success is the talent of human capital to create, use and improve technologies in the concept of a Smart City [48].

The primary goal of the digital development plan is to achieve citizen satisfaction with urban management technologies at the level of 75 to 80% (the goal was achieved in 2018). By meeting the goals, trust is built on the basis of trust-aware systems (T-ARS). The principles of the strategy form the core of digitization (data improves processes) and the provision of services for the needs of stakeholders (for a government that "serves with heart") [48].

The city has three stakeholders, i.e., citizens, business and the public sector. The output should be the adoption of modern and advanced technologies that [48]:

- are easy to use;
- meet the requirements of relevance, security, digital access;
- are responsible for data protection;
- allow seamless implementation and use.

In Singapore, technology is seen as an investment for development, not an expense. Effective digitization requires a link between technology and city policies through new organizational structures. The head of the digital strategy, who also coordinates the Information and Communication Technologies Committee, is in charge of meeting the set goals [48]. The digitization of the city is also supported by the Department of Communications and Information through four steps [47,48]:

- securing access to technologies and applications (via Service-Dominant Logic, Software-Defined Network, Wireless Sensor Network);
- creating awareness;
- community support;
- deliberately integrating digital elements and information and communication technologies into the daily lives of citizens, using the Smart Nation application.

Singapore distributes the collected field data to the Operational Management System model platform. In this way, data are processed into information and supports managerial functions of management and decision making [47].

The urban model consists of a sensory layer and platforms, middleware (testing, development, data analytics and monitoring), micro-services (e.g., National Digital Identity) and comprehensive digital services. The result of the model is improved user experience, quality Smart City management processes and simple implementation [48].

Singapore provides information on weather, the environment, water resources and the traffic situation in the city through information panels. It also uses a 5G MEC (mobile edge computing) network by Ericsson [49–51].

Smart City Singapore is a safe city; in 2018, there was no theft or robbery in the course of 322 days a year. The credit for this result is due to sensor cameras and the modern technology of Lamppost as a Platform, which uses software to recognize the characteristic features of the face. Police officers use virtual smart glasses (real-time face analysis) or drones in their city protection activities [47].

UAV (unmanned aerial vehicle) equipment is used in Singapore to inspect canals and structures (saving 60 to 80% of inspection costs), food delivery, mosquito replication, monitoring, security or rescue missions. In addition to classic drones, the so-called NUSwan, an unmanned aerial vehicle, takes the form of swans that assess drinking water quality [52,53].

Drones represent a trend and a popular hobby that has resulted in the creation of a community of drones in Singapore, the so-called SG UAS. Testing of new advanced versions of drones is carried out in the Seletar Aerospace Park technology park. Drones minimize personal contact, so they are also suitable for use in the current Covid-19 pandemic, for example in the form of crowd surveillance. Drones distinguished whether people wore a mask and kept a distance of 2 m. The limitations of current unmanned aerial vehicles are their weight and safety [53].

Singapore is addressing the issue of unmanned aerial vehicle safety by setting permitted runways, based on best practices in the US, Japan and South Korea. Each drone must be traceable and can only be registered by persons over 16 years of age [54].

As part of the management of limited resources, Smart City Singapore relies mainly on water. Monitoring takes place via the WaterWiSe platform. The input for the platform is data from sensors and IoT. The data are then integrated and sent to the electronic messaging system. The output is information on pressure, water quality and ruptured pipes. Measurements take place at 5 to 15 min intervals. The information is used to optimize flows, pipelines, predict water consumption and create simulation models. The benefits are efficient management and decision-making processes for the management of limited resources in Smart City Singapore [55].

4.1.2. London

The Smart City London strategy until 2018 includes five missions, the fulfillment of which is to bring customer-oriented services using the principle of customer service at the heart of the process, easy connectivity for all stakeholders, data leadership open to innovations in advanced technologies and strong collaboration [56].

London is one of the world leaders in the use of data collected from air protection sensors through the so-called Air Quality Network. The Wireless Sensor Network was built in collaboration with Imperial College London and Bloomberg Philanthropies. In 2019, 100 IoT sensors were implemented in London to monitor air quality [56,57]. IoT-based smart sensors are also used in strategic city management. The FlexLondon initiative supports new digital technologies for secure analysis and sharing of Big Data. Their storage is based on the London Datastore, which contains data from more than 700 datasets [56]. Within the network infrastructure, the Connected London program has been introduced, which is preparing the transition to a 5G network from Vodafone [58]. The London Plan provides all households with full fiber and mobile connectivity based on public Wi-Fi and a separate network for administration called Govroam [56].

The Met Police app maps criminal activity and promotes security in the Smart City. The data is used to prevent theft. The city has implemented 22,000 intelligent sensor cameras, which transmit data to dashboards. Data security is the responsibility of the Mayor's Office for Policing and Crime (MOPAC) and the London Resilience Group. In 2017, the London Digital Security Center was created, which uses the information security policy model [56]. The test center for open platforms is Queen Elisabeth Olympic Park. London is currently focusing on drone technology projects. Digital Greenwich is building new data standards and technologies to reduce the consumption of energy produced in buildings and public lighting. London Councils is working on the London Ventures project to foster cooperation between the private sector, the public sector and citizens [56].

The city management is implementing the Smart London Camp, where citizens have the opportunity to participate in a public debate and convey their views and expectations from Smart City projects. The inhabitants of the city care about the development of such advanced technologies, which at the same time build trust and transparency. Mayor Sadiq Khan wants to focus in the future on strengthening trust in systems and data through artificial intelligence [56].

Regular online communication is also implemented through the Talk London platform. Using dashboards and online platforms helps create new Smart City models. Thus, citizens are not only participants but also co-creators [56].

The primary goal of strategic management is to promote the security of shared data through education, linking technologies with the social element of culture, such as Smithfield Market or Digital Design Weekend events [56].

Drones are generally used in London in agriculture, health, safety, environmental protection and flood management. The control takes place via NESTA Flying High. Drone testing is carried out in rural areas with low population density. In 2021, drones were used to test the stability of a building, saving more than 80% of resources and time [59,60].

Since 2017, the research team in London has been focusing on the development of new drones that will find use as air and water pollution monitoring units. In 2019, the first experimental trials of a drone called "flying fish" were carried out, which can dive underwater and monitor data from the River Thames. An extended version is being developed in collaboration with Switzerland [61–63].

### 4.1.3. Helsinki

The Smart City framework of the second most successful city according to the Smart City Index 2020 consists of elements of strategy, technology, government and stakeholder management. The city's strategy is set for the period 2017–2021 with the primary goal of constantly improving the quality of life and producing zero emissions [64].

The city wants to provide residents with a safe and trustworthy space to live, thus supporting T-ARS (trust-aware systems). A critical factor for success is digitization through the adoption of modern and advanced technologies [64].

The core consists of a sensor network based on the Internet of Things, Service-Dominant Logic, Software-Defined Network and Wireless Sensor Network (IoT, SDL, SDN and WSN). Information and communication technologies and data management are in charge of the Economic Development and Planning Division. Statistical processing of Big data helps to model 3D models of the city. Modern technologies are being developed in Helsinki in cooperation with Forum Virium Helsinki for mobility, the environment and the city's economic development. The city provides services on the basis of SDL (Service-Dominant Logic) and the so-called Service Map (citizens search for services via a digital platform). The government supports all new Smart City projects which it finances, raises awareness of the issue and participates in the creation of Smart City concepts. It uses analyzed data in the form of information to support management and decision making. Stakeholders are driven by their needs and values. Their views, attitudes and values are collected through intelligent communication platforms in the city [64]. The technological architecture in Helsinki is called Snap4City. The sensors collect data from the fields of transport, healthcare, social networks and media, limited resources (especially air) and the government. Data are aggregated through IoT applications, then analyzed and used in city management processes [65].

The city of Helsinki publicly shares the collected data from the sensory network on city dashboards. In cooperation with Nokia, an MEC (mobile edge computing) network with object tracking, camera surveillance and video analysis applications has been used since 2016 [65,66].

Helsinki promotes data security through information and communication training programs; in an effort to reduce the amount of data shared, city lawyers increase the provision of advice and advise on personal data protection [64].

The primary preference for drones in the Smart City Helsinki area is focused on reducing emissions (flying taxis) and protecting a limited resource of air. Carbon neutral solutions are popular in Finland. Transport of an item from point A to point B often has lower emissions using an unmanned aerial vehicle (UAV) device. Forum Virium Helsinki offers citizens free courses and seminars on drones. Testing is carried out in municipalities that form experimental platforms. The government strongly supports unmanned aerial vehicle technology, as Helsinki has significant airspace. Legislation on drones is flexible and adaptable. Restrictions relate to safety and regulation. The challenge is not the technology itself but its acceptance by some conservative citizens [67,68].

In 2017, the University of Helsinki created the MegaSense program, which is used to monitor air quality based on 5G technology, sensors and the IoT network [69].

The collected data are evaluated with the Helsinki Region Environmental Services Authority. Statistics are further processed via the Snap4City architecture and shared on city dashboards and mobile applications [65].

*4.2. Comparative Analysis*

All three analyzed Smart Cities (Singapore, London and Helsinki) have advanced IoT technology implemented in their concepts. Singapore uses a sensor and communication platform to analyze Big Data. In London, shared data are stored in the London Datastore. Data are then analyzed and used to support management and decision making. Smart City Helsinki manages data through a separately created division of Economic development and planning (Sections 4.1.1–4.1.3).

To build and manage Smart City concepts, it is essential to create an environment based on the trust of citizens. Singapore aims to "serves with heart", trying to reflect the needs and expectations of its people. The Talk London platform serves as a communication tool through which citizens communicate their views and comments. Once fulfilled, trust between all parties involved increases. To promote trust, Helsinki uses the services of an environmental authority, which provides citizens with relevant and transparent information (Sections 4.1.1–4.1.3).

Service-Dominant Logic in Singapore is primarily focused on citizens and easy administration of activities such as payments or mobility. In London, the SDL focus is on supporting connectivity and development plans. Helsinki has a separately created Service Map. A common element is the primary focus on people and their needs (Sections 4.1.1–4.1.3).

All analyzed cities use a Wireless Sensor Network (WSN). The Software-defined network element has been applied in Singapore in a similar way to confidence support systems ("serves with heart"). The London network uses the area for cooperation through the London Ventures project to achieve sustainable goals. Helsinki prefers the use of SDN mainly due to the implementation of intelligent communication platforms (Sections 4.1.1–4.1.3).

A common features of specific urban platforms in Singapore, London and Helsinki are centrist citizen orientation, data collection, analysis, management and decision support. Cities publish the information obtained transparently on their websites and information panels, thus increasing the awareness, trust and participation of stakeholders (Sections 4.1.1–4.1.3).

To support data sharing in Smart City concepts, the analyzed cities use mobile edge computing technology. In Singapore, it is provided by Ericsson, in London via Vodafone and in Helsinki via Nokia (Sections 4.1.1–4.1.3).

In the current era of cyber attacks, data sharing and increased crime in populated cities, residents are worried about their security. Singapore's strategic management solves the problem through surveillance systems, such as drones. Groups and centers have been set up in London to share data securely online. As part of physical security, police officers use new advanced technologies to detect the perpetrator and maintain public order. In Helsinki, management addresses safety issues by supporting education through training programs and data minimization (Sections 4.1.1–4.1.3).

The current trend in the field of advanced technologies in practice is the drones that Singapore, London and Helsinki use for monitoring activity in terms of security or monitoring the consumption of limited resources. Each of the analyzed cities uses specific standards for drone flight operations. In Singapore and London, standards are set by the public sphere, in Helsinki this is done by the private company Forum Virium Helsinki. Singapore prefers monitoring water resources, Helsinki specializes in air and London focuses on both items, water and air (Sections 4.1.1–4.1.3). A summary of compared advanced technologies is provided in Table 5.

**Table 5.** Summary of compared advanced technologies in Smart Cities Singapore, London and Helsinki.

| Advanced Technologies | Practice | | |
|---|---|---|---|
| Theory | Singapore | London | Helsinki |
| Internet of Things (IoT) | Yes | Yes | Yes |
| Big data | Sensor and Communication platform | London Datastore | Economic Development and Planning Division |
| Trust-aware systems | Serves with heart | Talk London | Helsinki Region Environmental Services Authority |
| Service-Dominant Logic | National Digital Identity, Smart Urban Mobility, CODEX, Unique Entity Number, Smart Nation, The Moment of Life Initiative | Connected London, Govroam, London Plan | Service Map |
| Software-Defined Network | Serves with heart | 5 Missions, London Ventures | Intelligent communication platforms |
| Wireless Sensor Network | Yes | Yes | Yes |
| Urban model of access platform | Operational Management System | Talk London | Snap4City |
| Dashboards | Yes | Yes | Yes |
| Mobile edge computing | Ericsson | Vodafone | Nokia |
| Information security policy | Lamppost as a Platform, smart glasses, drones | The Met Police, FlexLondon, Mayor's Office for Policing and Crime, London Resilience Group, London Security Center | Training programs, professional consultations, data minimization |
| Unmanned aerial vehicle | SG UAS | NESTA Flying High | Forum Virium Helsinki |
| Sensors for environmental monitoring (water, air) | NUSwan, drones, WaterWiSe (water) | Drones, "flying fish" (air, water) | MegaSense, drones (air) |

Source: own processing according to [47–69].

*4.3. Key Findings*

The secondary research confirmed that all compared Smart Cities use a set of 12 technologies identified in the theoretical section of the article (Sections 2.1–2.7). Table 5 summarizes the use of specific programs, technologies, systems or responsible actors from Section 4, commonly used elements are marked as "yes", which confirms their use in practice.

Strategic management of best practice cities adapts technologies to the needs and requirements of its citizens, thus connecting the technological aspect with the managerial and social ones. The basic use of the identified advanced technologies has the same use in the practices of cities.

The specific elements are mainly in the preference for monitoring a particular limited resource, which is related to local conditions. Singapore and London provide their solutions in cooperation with the government and public institutions, while Helsinki prefers cooperation with the primary sector. However, the use of the technological basis has the same goal in all analyzed cities, meeting the needs of citizens as much as possible. In this way, Singapore, London and Helsinki contribute to the creation of centrist-oriented Smart City concepts.

**5. Discussion**

Based on the secondary analysis of the literature (Section 2) and the results of the summarization and comparison (Section 4.2, Table 5), it was possible to answer the second research question as follows:

- How is it appropriate to implement advanced technologies for Smart City management in general?

Elements of the general model of implementation of advanced technologies for Smart City management (Figure 1) were selected based on a set of 12 common technologies identified according to Section 2 and applied in practice according to Section 4.

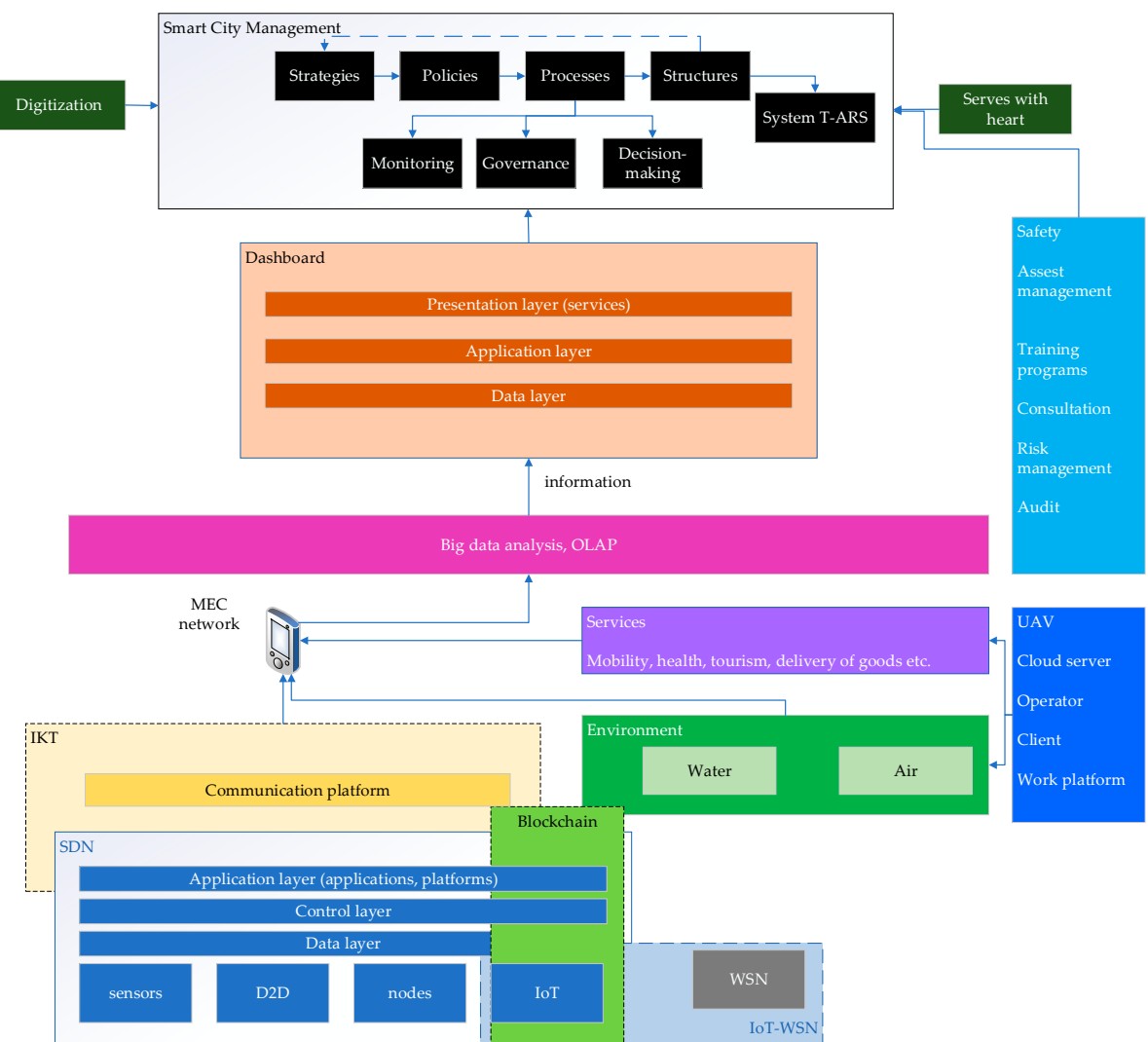

**Figure 1.** General model of implementation of advanced technologies for Smart City management (own processing by authors according to the results of literature review in Sections 2.1–2.7 and case studies in Section 4).

### 5.1. Elements for Model Creation Obtained from the Theoretical Part of the Article

The basic layer of the model in Figure 1 is SDN (Software-Defined Network) technology, which according to Rahauti, Xiong, Xin has three layers, i.e., data (sensors, device-to-device, nodes and IoT), control and application (applications and platforms). Wireless Sensor Network (WSN) technology overlaps with the Software-Defined Network (SDN) layer through the use of Internet of Things technology, which, according to Kedhr et al., creates the IoT-WSN network (Section 2.3).

At the same time, SDN overlaps with information communication technologies, thus generating communication platforms. According to Iftikhar et al., Conway and Kundu, the joint use of IoT and blockchain has a positive impact on environmental monitoring (Section 2.7).

According to Giyenko, Cho, the architecture of an unmanned aerial vehicle (UAV) has four elements (cloud server, operator, client and work platform in Section 2.6). Drones affect environmental monitoring and the layer of services in the field of transport, health, tourism, transport of goods, etc.

According to the authors of the article, communication platforms, the environment and services in the MEC (mobile edge computing) network are interconnected. The collected

data from the MEC network are transferred to the Big Data analysis layer, OLAP (Online Analytical Processing). The output is information that is published via a dashboard.

Farmanbar and Rong (Section 2.4) describe the dashboard architecture with three layers: data, application and presentation (for services). The authors are of the opinion that citizens can effectively participate in Smart City decision-making processes through free access to information via dashboards.

According to the authors, the safety aspect is a critical element of the success of the implementation of advanced technologies for Smart City management (Section 2.5). Wu, Su and Wu are of the opinion that security needs to be maintained through asset management, risk and audit.

The authors are of the opinion that relevant and reliable information has a positive effect on the creation of strategies, policies, processes (monitoring, management and decision making) and organizational structures, thus supporting trust-based systems, i.e., T-ARS (trust-aware systems) (Section 2.3).

*5.2. Elements for Model Creation Obtained from the Analysis of Case Studies*

Based on the findings from Section 4, i.e., analysis of case studies, specific elements were used in the model (Table 6).

**Table 6.** Specific elements characteristic for the three analyzed cities—Singapore, London and Helsinki.

| Specific Element | Smart City | | |
| --- | --- | --- | --- |
| | **Singapore** | **London** | **Helsinki** |
| Monitoring of water resources through UAV | x | x | |
| Air monitoring via UAV | | x | x |
| Serves with heart | x | x | x |
| T-ARS | x | x | x |
| Safety | x | x | x |
| Digitization strategies | x | x | x |
| Training programs | | | x |

Source: own processing according to Section 4.1.1, Section 4.1.2, Section 4.1.3, Section 4.2. and Table 5.

It is typical for Singapore to use drones in the monitoring of a limited water source. It specifically uses advanced digital technologies to support and build trust in the form of T-ARS systems and services called Serves with heart.

London focuses on water monitoring and air protection. Specific elements are the strong emphasis on future strategies, citizens' trust and security in the online and physical worlds.

Helsinki prefers monitoring and air protection. The city addresses security through elements of education, implementation of training programs and training from the primary sector. Helsinki promotes an environment of mutual trust and co-development by promoting security and sharing transparent information.

All these significant elements summarized in Table 6 form part of the general model itself in Figure 1. The common elements in Table 6 are marked with a cross.

Citizens in the analyzed cities fulfill not only the function of technology users, but also participants and co-creators in the creation of strategies.

Singapore, London and Helsinki use advanced technologies based on the needs of citizens (according to main findings in Table 6), which can be summarized with the goal of "serving with heart". By conducting discussions with the population, they adjust not only their strategic goals, but also their technological possibilities. Citizens in all three cities thus act as users of advanced technologies, participants and co-creators (according to Table 6 and Simonofski et al.) The basic finding of the summary and data comparison from case studies is to confirm the connection between theory and practice and to emphasize the impact of technology on the human factor of cities. The authors incorporated the link

between digitization and the social aspect of technology ("serves with heart") as influences that act on the strategic layer of the model in Figure 1. The authors used these data for design in the Strategic City Management section, which is also affected by dashboard information security. Strategic city management should create goals and projects on the basis of secure and transparent information in cooperation with citizens, thus supporting the effective creation, development and stabilization of Smart City management concepts.

*5.3. Limitations and Applicability of the General Model*

The limitations of research were formed by national and cultural differences of cities. A limitation is also the adaptation of the model to the specific conditions of the selected city or the absence of its testing, which is planned in the future.

The proposed model is planned to be applied in the future for a selected Slovak city, which will meet the preconditions to become the new Smart City.

The limitations of the article include a logical reduction of articles from two databases (Web of Science, Scopus) according to the specified keywords and a detailed analysis of the most relevant ones. Another limitation is the selection of case studies, which include one city from Asia (Singapore) and two from Europe (London and Helsinki).

The results of the article provide fellow researchers with the findings of a secondary analysis of relevant articles, from which they can draw when writing their own publications without the need for time-consuming search of the articles about this topic in databases.

The general model of implementation of advanced technologies serves as a basis for strategic management of new Smart Cities that want to implement a technological base and at the same time do not want to forget the managerial and social aspects.

## 6. Conclusions and Recommendations

Effectively managed Smart Cities based on the synergy between the managerial and technological aspects reach the leading positions in Smart City rankings and global indices. The management of the world's Smart Cities is significantly influenced by the 12 identified advanced technologies, including:

- Internet of Things (IoT);
- Big Data;
- T-ARS system (trust-aware systems);
- SDL (Service-Dominant Logic);
- SDN (Software-Defined Network);
- WSN (Wireless Sensor Network);
- MEPA model (urban model of the access platform);
- Dashboards;
- MEC (mobile edge computing) network;
- ISP (information security policy) security model;
- UAV equipment (unmanned aerial vehicle);
- sensors for environmental monitoring.

Singapore, London and Helsinki also use all identified advanced technologies in their practice, taking into account the needs of citizens. Based on the acquired knowledge, the authors recommend to the strategic management of the newly emerging Smart Cities to build a strong technological infrastructure, which must reflect the needs and expectations of citizens, by not only collecting data but also analyzing and using them to support management and decision-making.

Advanced technologies have great importance for the future for the management of Smart City processes, resources and stakeholders. Cities should see advanced technologies as a future trend in a holistic way of solving social and environmental problems. The interconnection of all layers of the own model will ensure the efficient operation and management of Smart City concepts of the present time. Implementation and validation of the model in practice forms the basis for further research activities of the authors in the future.

**Author Contributions:** Conceptualization, D.Š., J.V. and M.K.; supervision, J.V. and M.K.; formal analysis, D.Š.; methodology, D.Š.; writing—original draft, D.Š. All authors contributed to the manuscript preparation. All authors have read and agreed to the published version of the manuscript.

**Funding:** This article was realized with support of the Operational Programme Integrated Infrastructure in frame of the project: Intelligent systems for UAV real-time operation and data processing, code ITMS2014+: 313011V422 and co-financed by the European Regional Development Found.

**Institutional Review Board Statement:** Not applicable.

**Informed Consent Statement:** Not applicable.

**Data Availability Statement:** No new data were created or analyzed in this study. Data sharing is not applicable to this article.

**Conflicts of Interest:** The authors declare no conflict of interest.

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
