# Peer review of "Advanced Technologies and Their Use in Smart City Management"

_sustainability, doi:10.3390/su13105746_

Round 1

Reviewer 1 Report

The research paper aims was to defined to propose a general model for the implementation of advanced technologies for Smart City management. The main goal of the research, it is necessary to obtain answers to two research questions: What advanced technologies are currently used to manage Smart City? and How is it appropriate to implement advanced technologies for Smart City management in general? 

According to the reviewer, the purpose of the study was achieved, but there are questions:

  1. Please consider whether using only the database  Web of Sciene gives a comprehensive view of the problem under study. Are the results representative if we use only one database? What about other knowledge bases ?
  2. Why was the comparative analysis performed for 2 cities from the proposed list SCI 2020, and not for e.g. 3 or 5 cities?

Please consider my suggestions. 

Good luck and health in difficult times of the pandemic !!!

Reviewer 2 Report

I believe that the paper is really interesting however because it is a case study I recommend the authors to follow a more structural way to moderate and present both case studies. Though I would recon more than 2 case studies to be compared in order to review the work in a more holistic perspective and the comparison to be objectively and more scientifcally acceptable.

I recommend you follow https://www.monash.edu/rlo/quick-study-guides/writing-a-case-study

This will help you to reshape the paper in a more academic basis.

Further to this the reseach methodology is missing following by relavant justification.

Reviewer 3 Report

This paper aims to suggest a general model for the implementation of advanced technologies for Smart City management through a literature review and case studies.

The overall objective of the paper is interesting but several elements make it difficult for me to see it published as-is:

- The writing of the paper can be largely improved. Just in the abstract, there are incomprehensible formulations (“Building Smart City management concepts is based on the implementation and use of 8 advanced technologies”) or wrong formulations (“Smart Cities uses 12 identified general advanced technologies”). I would urge the authors to perform a re-reading and rewriting of the paper.

- The two research questions are not clear nor linked with the literature. They seem to appear out of nowhere after a very brief introduction. What is the motivation for these questions ? I would suggest to the authors to better position them with the existing knowledge base.

- The paper presents several technologies but it is never justified why these technologies are presented. I think that, as it constitutes results, it should be move in the results section after the reader has gained knowledge about the methodology. Furthermore, all the bullet points to present the technologies make it very difficult to read.

- The rationale behind the case studies is well-explained but I would have liked more information about the limitations of using this specific index to select the case studies, as one index only focuses on some criteria but could miss on other. For instance, I found that the participatory aspect of smart cities was not enough discussed in the paper with a focus only on the technological aspect (for more info, see: Simonofski, A., Serral Asensio, E., De Smedt, J. et al. Hearing the Voice of Citizens in Smart City Design: The CitiVoice Framework. Bus Inf Syst Eng 61, 665–678 (2019).)

- Figure 1 constitutes a core contribution but, for now, the traceability between the model and the findings is not clear. The authors mention the use of deduction but more information would be welcome.

- The discussion fails short in highlighting the relevance of the paper for fellow researchers and for practitioners. To who the results will be helpful ? How can practitioners use the general model?

Reviewer 4 Report

In this contribution, the authors discuss about the advanced technologies to manage Smart Cities, using Helsinki and Singapore as two case studies. In general, manuscript is a well written, and this is useful MS for the sub domain of Smart Cities. However, I have two major concerns for this paper, and my recommendation of acceptance is conditioned on if the authors can address my concerns:   1. Currently, the manuscript does not flow very well. I'd like the authors to emphasize or highlight the research questions they answer (e.g., line 51-52), and do a better job at summarizing the recommendations (which is a bit verbose right now).   2. Please do a better literature review. It should be better to cite one or two paper per use case of IoT in Section 1.1, instead of citing one review in total. As an example, the reviewer recommend to cite the following paper in Section 1.1, for people as a sensor category: (DOI: 10.1109/SmartCity.2015.54).   Following are minor edits: 1. Note: fix the resolution in Figure 1. 2. English can be improved. Please do a thorough round of proofreading and improve the manuscript quality.  

Round 2

Reviewer 2 Report

The paper has to be better structured to fulfill Introduction, Research Background, Research Methodology, Data Presentation, Findings and Analysis, Conclusions, Recommendations The findings from the Case Study analysis are not presented. Learning outcomes? What did you learnt and how did you shape the framework? Is the framework validated or not? Please explain and justify your answers.

Reviewer 3 Report

Dear authors, 

Thank you for the changes made.  They address my comments. My only remaining suggestions would be to perform an overall proof-reading of the paper to improve its readability and scientific writing. 

Reviewer 4 Report

After the revision, I believe that the manuscript is more complete.
